# Performance Comparison of Meta-Heuristics Applied to Optimal Signal Design for Parameter Identification

**DOI:** 10.3390/s23229085

**Published:** 2023-11-10

**Authors:** Accacio Ferreira dos Santos Neto, Murillo Ferreira dos Santos, Mathaus Ferreira da Silva, Leonardo de Mello Honório, Edimar José de Oliveira, Edvaldo Soares Araújo Neto

**Affiliations:** 1Department of Electroelectronics, Federal Center of Technological Education of Minas Gerais (CEFET-MG), Leopoldina 36700-001, Brazil; murillo.ferreira@cefetmg.br; 2Faculty of Engineering, Federal University of Juiz de Fora (UFJF), Juiz de Fora 36036-900, Brazil; mathaus.silva@engenharia.ufjf.br (M.F.d.S.); leonardo.honorio@ufjf.edu.br (L.d.M.H.); edimar.oliveira@ufjf.br (E.J.d.O.); 3Santo Antônio S.A., Hydroelectric Plant Santo Antônio, Porto Velho 76805-812, Brazil; edvaldoneto@santoantonioenergia.com.br

**Keywords:** Optimal Signal Design, parametric estimation, meta-heuristics, Autonomous Surface Vehicles

## Abstract

This paper presents a comparative study that explores the performance of various meta-heuristics employed for Optimal Signal Design, specifically focusing on estimating parameters in nonlinear systems. The study introduces the Robust Sub-Optimal Excitation Signal Generation and Optimal Parameter Estimation (rSOESGOPE) methodology, which is originally derived from the well-known Particle Swarm Optimization (PSO) algorithm. Through a real-life case study involving an Autonomous Surface Vessel (ASV) equipped with three Degrees of Freedom (DoFs) and an aerial holonomic propulsion system, the effectiveness of different meta-heuristics is thoroughly evaluated. By conducting an in-depth analysis and comparison of the obtained results from the diverse meta-heuristics, this study offers valuable insights for selecting the most suitable optimization technique for parameter estimation in nonlinear systems. Researchers and experimental tests in the field can benefit from the comprehensive examination of these techniques, aiding them in making informed decisions about the optimal approach for optimizing parameter estimation in nonlinear systems.

## 1. Introduction

The design of input signals is undoubtedly one of the key elements for the success of system identification [1]. Well-designed signals can stimulate and highlight the main dynamics of the system, allowing these behaviors to be accurately estimated and represented by parametric models [2]. Consequently, they constitute a fundamental element for obtaining accurate models and, in turn, for tuning model-based controllers [3], designing state observers [4], and developing simulators [5], among other applications [6].

However, the task of Optimal Signal Design (OSD) is not a trivial problem. As presented in the literature, especially in [7,8,9], this task is defined as a mixed-integer nonlinear optimization problem. It requires competent approaches to simultaneously address signal design and parametric estimation problems.

Regardless of the methodology adopted, there is a consensus that the approach must generate signals that provide effective stimuli to the main dynamics of the system [10]. These signals must exhibit appropriate excitation persistence, covering a wide range of amplitudes and frequencies. Furthermore, it is essential that the signals be feasible for the actuator system and adhere to safety conditions during the identification experiment development [11].

Meeting all these requirements is challenging, especially in nonlinear systems, where signals with high persistence and detailed encoding are necessary. In such cases, there can be a significant increase in computational complexity, leading to time-consuming computational solutions [12].

To address the Optimal Signal Design (OSD) problem, it is common to formulate the optimization problem using the inverse of the Fisher Information Matrix (FIM). This approach allows for measuring signal excitation persistence, irrespective of the type of estimation algorithm implemented.

Regarding the solution to the optimization problem, various deterministic and probabilistic optimization algorithms are employed [13]. These include Steepest-descent, Conjugate-gradient, Quasi-linearization, and Newton–Raphson, among others. Additionally, bio-inspired meta-heuristics are classic examples of probabilistic methods [14].

A comprehensive survey of the literature reveals several algorithms in this context. Concerning solutions based on deterministic methods, works such as [15,16] serve as classic examples of applications in the field of OSD. In these approaches, signals are optimized using deterministic algorithms that target metrics based on the Fisher Information Matrix (FIM), including criteria like A-Optimality, E-Optimality, and D-Optimality

Another approach found in the literature combines deterministic methods with optimal control theory. Examples of these combined approaches can be found in [17,18,19,20].

Specifically, in the work by [17], an identification methodology is applied to aerial vehicles. The optimal signals are generated through a combination of dynamic programming and optimal control, which enhances traditional identification maneuvers. It is worth noting the significant reliance on the optimal control input signal. While these approaches yield satisfactory results, it is evident that further improvements to the entry maneuvers could lead to a substantial increase in the problem’s spatial complexity, rendering it intractable for the techniques presented.

In terms of methodologies, it is worth highlighting the works of [7,12,21], which focus on designing multilevel pseudo-random sequence signals. The work in [7] employed the PSO algorithm, ref. [12] utilized the Ant Colony Optimization (ACO) algorithm, and [21] opted for the Genetic Algorithm (GA) to address the OSD problem. What sets these works apart is their ability to design signals with advanced coding, reasonable processing times, and high-quality solutions. In the case of [7], the signal design also optimized the experiment time, a feature not found in other works. Additionally, the authors developed a dual-layer optimization methodology, using their own metrics that cater to specific needs.

Considering all the mentioned works, it becomes apparent that Optimal Signal Design can be highly complex, particularly when dealing with signals featuring multiple levels of amplitude or intricate coding. In such cases, the number of potential combinations in the solution space can become prohibitively large, potentially rendering the project infeasible or leading to extremely high computational processing times, as exemplified in [12].

In this context, the application of meta-heuristics has shown promise, as demonstrated in the works of [7,12,21]. However, a more comprehensive investigation, involving a comparison of various techniques, is still needed. It is worth noting that this area of research is not yet extensive in the literature, but it holds promise for OSD and deserves further attention.

In light of this context, this paper presents a comparative study of meta-heuristics applied to the OSD problem for estimating parameters of Nonlinear Dynamic Systems (NDSs). Several meta-heuristics will be investigated using the methodology proposed by [22], referred to as rSOESGOPE. While the PSO algorithm [23] served as the optimizer, this paper will also explore other meta-heuristics to enhance the performance of rSOESGOPE. The traditional algorithms such as GA [24], Bat Algorithm (BA) [25], Whale Optimization Algorithm (WOA) [26], Grey Wolf Optimizer (GWO) [27], Salp Swarm Algorithm (SSA) [28], Arithmetic Optimization Algorithm (AOA) [29], and Multi-trial vector-based Differential Evolution (MTDE) [30] will be analyzed.

This comparative study aims to explore and assess the potential of these approaches for solving this class of optimization problems. To this end, real case studies involving an ASV with three DoFs and an aerial holonomic propulsion system are presented and discussed.

After this summary, the contributions of this paper can be highlighted as follows:a performance comparison of eight meta-heuristics applied in the rSOESGOPE concept for parametric estimation, not yet explored in the literature;a description of the rSOESGOPE concept that allows obtaining robust models from the use of multiple projection excitation signals;a real case application involving a 3 DoF ASV, which presents a problem with a complex solution space for signal generation and a challenging parametric estimate due to the hydrodynamic phenomena involved;an in-depth analysis of the ASV’s identification process, providing valuable insights into the modeling and generation of identification signals.

This paper is organized as follows: Section 2 presents the OSD problem statement, demonstrating its characteristics and complexities; Section 3 introduces the rSOESGOPE method; Section 4 elaborates on the fundamental aspects for the application of the algorithms in the rSOESGOPE concept; Section 5 presents the results of applying the meta-heuristics to a real case study problem; Section 7 presents some concluding remarks.

## 2. Problem Formulation

Consider a nonlinear dynamic system R(Γ) which can be satisfactorily approximated by the nonlinear model M(Γ):(1)$$\begin{array}{c} M(\Gamma ):=\left\{\begin{array}{c} \dot{x}\left(t\right)=f\left(x\left(t\right),u\left(t\right),\Gamma \right)\\ y(t)=h(x(t),u(t),\Gamma ) \end{array}\right., \end{array}$$
where f and h are the system-dependent nonlinear functions on the state vector $x\in R^{n}$, the output vector $y\in R^{m}$, the input vector $u\in R^{p}$, and the set of parameters of the model $\Gamma \in R^{r}$. Additionally, it is admitted that the R(Γ) has some restrictions that need to be respected during its operation, represented by
(2)$$\begin{array}{c} \begin{array}{c} \underline{x}\leq x\left(t\right)\leq \bar{x},\\ \underline{y}\leq y\left(t\right)\leq \bar{y},\\ \underline{u}\leq u\left(t\right)\leq \bar{u}, \end{array} \end{array}$$
where $\underline{x}$ and $\bar{x}\in R^{n}$ represent, respectively, the upper and lower vector states’ operating limits, $\underline{y}$ and $\bar{y}\in R^{m}$ express the lower and upper vessels’ output vector operating limits, respectively, while $\underline{u}$ and $\bar{u}\in R^{p}$ represent the upper and lower input signal operating limits, respectively.

It is also assumed that Γ represents the unknown optimal set of the system’s parameters, and M(Γ) best represents R(Γ) for any given signal u∈U, where U represents the entire possible input signal domain.

In this context, designing an optimal identification signal involves finding the best signal $u^{⊕}\in U$ that, when applied to the real system R(Γ), enables the subsequent parametric estimation ${\hat{\Gamma }}^{+}$, satisfying the following relationship: $M({\hat{\Gamma }}^{+},u)\approx R(\Gamma ,u)$. Therefore, designing an optimal identification signal requires discovering $u^{⊕}$ through an optimization search defined by
(3)$$S\left(M\left({\hat{\Gamma }}^{-}\right),R\left(\Gamma \right)\right)=u^{⊕},$$
where ${\hat{\Gamma }}^{-}$ is the initial parameter estimation ${\hat{\Gamma }}^{-}$ and S is an arbitrary mixed-integer optimization algorithm that searches for the best $u^{⊕}\in U$. Finally, the optimal parameter set is defined by
(4)$$P\left(M\left({\hat{\Gamma }}^{-},u^{⊕}\right),R\left(\Gamma ,u^{⊕}\right)\right)={\hat{\Gamma }}^{+},$$
where P is an arbitrary optimization procedure that uses the mathematical model M and the signal $u^{⊕}$ to estimate the parameter set ${\hat{\Gamma }}^{+}$ in which $M({\hat{\Gamma }}^{+},u)$ best represents $R(\Gamma ,u^{⊕})$. Additionally, in this notation, it is possible to replace [x,y] by M(Γ,u) or R(Γ,u) without loss of generalization.

## 3. Robust SOESGOPE

The Robust SOESGOPE (rSOESGOPE) is a methodology developed for designing identification signals and promoting robust parametric estimation.

This approach is a derivation of the original SOESGOPE method developed by [7] for designing single optimal signals for optimal parametric estimation of NDS.

However, as demonstrated by [8], the SOESGOPE method does not yield satisfactory results in situations where the initial estimate has a high level of uncertainty regarding the real system.

To address these uncertainties, the rSOESGOPE method proposes the use of multiple identification signals $U^{⊕}=\left[{u^{⊕}}_{1},{u^{⊕}}_{2},\ldots ,{u^{⊕}}_{n}\right]$ designed from a set of well-spatially distributed benchmark parameters $P^{⊕}=\left[{{\tilde{\Gamma }}^{p}}_{1},{{\tilde{\Gamma }}^{p}}_{2},\ldots ,{{\tilde{\Gamma }}^{p}}_{n}\right]$ around the initial estimation ${\hat{\Gamma }}^{-}$. With this change, the aim is to minimize uncertainties about ${\hat{\Gamma }}^{-}$ and enhance excitation persistence through multiple signals projected by different characteristics derived from the reference parameters. Consequently, this provides a more robust model for parametric uncertainties.

Mathematically, this new concept can be succinctly described by the following hypothesis:


*“If the set ${\hat{\Gamma }}^{-}$ is a rough approximation of Γ, where $P^{⊕}=\left[{{\tilde{\Gamma }}^{p}}_{1},{{\tilde{\Gamma }}^{p}}_{2},\ldots ,{{\tilde{\Gamma }}^{p}}_{n}\right]$ represents a set of well-spatially distributed benchmark parameters around the initial estimation ${\hat{\Gamma }}^{-}$, and $U^{⊕}=\left[{u^{⊕}}_{1},{u^{⊕}}_{2},\ldots ,{u^{⊕}}_{n}\right]$ leads to a set of signals capable of exciting $M(P^{⊕})$ and estimating $P^{⊕}=\left[{{\tilde{\Gamma }}^{p}}_{1},{{\tilde{\Gamma }}^{p}}_{2},\ldots ,{{\tilde{\Gamma }}^{p}}_{n}\right]$. Then, the same $U^{⊕}$ will also be able to excite R(Γ) and consequently obtain ${\hat{\Gamma }}^{+}$ that respects $M({\hat{\Gamma }}^{+},U^{⊕})\approx R(\Gamma ,U^{⊕})$.”*


These concepts can be better understood in Figure 1, which illustrates the trust region surrounding Γ, the designed signals, and the stages of parametric estimation.

Thinking in stages, the rSOESGOPE method proposes a two-step approach:Generate the benchmark parameter sets $P^{⊕}=\left[{{\tilde{\Gamma }}^{p}}_{1},{{\tilde{\Gamma }}^{p}}_{2},\ldots ,{{\tilde{\Gamma }}^{p}}_{n}\right]$ and use it as a parameter to find each signal of $U^{⊕}=\left[{u^{⊕}}_{1},{u^{⊕}}_{2},\ldots ,{u^{⊕}}_{n}\right]$.Apply $U^{⊕}$ to the real system to find ${\hat{\Gamma }}^{+}$.

Thus, according to Section 2, the following steps can be defined:(5)$$\begin{array}{ccc} U^{⊕} & = & S(M\left({\hat{\Gamma }}^{-}\right),M\left(P^{⊕}\right)), \end{array}$$
(6)$$\begin{array}{ccc} {\hat{\Gamma }}^{+} & = & P(M\left({\hat{\Gamma }}^{-},U^{⊕}\right),R\left(\Gamma ,U^{⊕}\right)). \end{array}$$

To tackle the two-step approach, a two-layer optimization strategy was developed to solve S. Two well-known optimization algorithms were employed: the PSO algorithm [31] and the Interior-Points Algorithm (IPA) [32].

The PSO algorithm was dedicated to the external optimization layer, specifically for the design of u(Ξ) and, consequently, the parameterization of Ξ, which represents the swarm’s individuals.

In the proposed method, the projected signals are AutoRegressive exogenous inputs with an Amplitude-Modulated Pseudo-Random Binary Signal (APRBS). The internal layer is assigned to the IPA, which is responsible for the parametric estimation required for evaluating signal quality. This same IPA is used to solve P and find ${\hat{\Gamma }}^{+}$.

To assess the signal quality generated by S, an objective function is proposed, consisting of a weighted sum of three metrics.

To ensure the analysis of excitation persistence, the following procedure was employed:PRECISION OUTPUT METRIC $f_{o}\left(\cdot \right)$: This metric is responsible for measuring the difference between the states and outputs obtained from $M({\hat{\Gamma }}^{+},u)$ and $M({\tilde{\Gamma }}^{p},u)$. It can be mathematically described as
(7)$$\begin{array}{c} J=\sum_{k=1}^{N}∥{\hat{x}}^{+}\left(k\right)-{\tilde{x}}^{p}\left(k\right)∥+∥{\hat{y}}^{+}\left(k\right)-{\tilde{y}}^{p}\left(k\right)∥, \end{array}$$
or, in the case of variables with significantly different dimensions,
(8)$$\begin{array}{c} J=\frac{1}{N}\sum_{k=1}^{N}{\left[{\hat{x}}^{+}\left(k\right)-{\tilde{x}}^{p}\left(k\right)\right]}^{T}Q_{x}\left[{\hat{x}}^{+}\left(k\right)-{\tilde{x}}^{p}\left(k\right)\right]+\\ \frac{1}{N}\sum_{k=1}^{N}{\left[{\hat{y}}^{+}\left(k\right)-{\tilde{y}}^{p}\left(k\right)\right]}^{T}Q_{y}\left[{\hat{y}}^{+}\left(k\right)-{\tilde{y}}^{p}\left(k\right)\right], \end{array}$$
where $\left({\hat{x}}^{+}\left(k\right),{\hat{y}}^{+}\left(k\right)\right)$ and $\left({\tilde{x}}^{p}\left(k\right),{\tilde{y}}^{p}\left(k\right)\right)$ are the states and outputs of $M({\hat{\Gamma }}^{+})$ and $M({\tilde{\Gamma }}^{p})$, respectively. *N* represents the number of samples *k* in the experiment ($t_{u}$), and $Q_{x}$ and $Q_{y}$ are weightings of the state and output variables, respectively. These weightings allow for the treatment of variables based on their degrees of uncertainty.RECOVERABILITY METRIC $f_{\hat{\delta }}\left(\cdot \right)$: This metric measures the capability to estimate the correct parameter set given u and ${\tilde{\Gamma }}^{p}$. Mathematically, $f_{\hat{\delta }}\left(\cdot \right)$ is represented by the sum of the relative errors between the final estimation ${\hat{\Gamma }}^{+}$ and the reference ${\tilde{\Gamma }}^{p}$:
(9)$$\begin{array}{c} f_{\hat{\delta }}\left({\hat{\Gamma }}^{+},{\tilde{\Gamma }}^{p}|f_{o}\right)=\sum_{i=1}^{r}\frac{|{\hat{\Gamma }}_{i}^{+}-{\tilde{\Gamma }}_{i}^{p}|}{|{\tilde{\Gamma }}_{i}^{p}|}, \end{array}$$
where ${\hat{\Gamma }}_{i}^{+}$ and ${\tilde{\Gamma }}_{i}^{p}$ represent the *i*-th parameter of ${\hat{\Gamma }}^{+}$ and ${\tilde{\Gamma }}^{p}$, respectively. This metric evaluates the similarity given the results provided by the previous metric $f_{o}\left(\cdot \right)$.It is important to mention that different parameter sets can generate the same output, and therefore, this metric also assesses how closely the estimated parameters match the benchmark set (${\tilde{\Gamma }}^{p}$).

To ensure compliance with the constraints, Θ(·) was used to penalize a given signal u(Ξ) if it drives the system out of the desired operational restrictions. Mathematically, this can be expressed as follows:(10)$$\begin{array}{c} \Theta \left({\tilde{x}}^{p},{\tilde{y}}^{p}|\Xi \right)=\left\{\begin{array}{c} \sum_{k=1}^{N}\sum_{i=1}^{n}\frac{|{\tilde{x}}_{i}^{p}\left(k\right)-\bar{x_{i}}|}{|\bar{x_{i}}|},\forall {\tilde{x}}_{i}^{p}\left(k\right)>\bar{x_{i}}\\ +\\ \sum_{k=1}^{N}\sum_{i=1}^{n}\frac{|{\tilde{x}}_{i}^{p}\left(k\right)-\underline{x_{i}}|}{|\underline{x_{i}}|},\forall {\tilde{x}}_{i}^{p}\left(k\right)<\underline{x_{i}}\\ +\\ \sum_{k=1}^{N}\sum_{i=1}^{m}\frac{|{\tilde{y}}_{i}^{p}\left(k\right)-\bar{y_{i}}|}{|\bar{y_{i}}|},\forall {\tilde{y}}_{i}^{p}<\bar{y_{i}}\\ +\\ \sum_{k=1}^{N}\sum_{i=1}^{m}\frac{|{\tilde{y}}_{i}^{p}\left(k\right)-\underline{y_{i}}|}{|\underline{y_{i}}|},\forall {\tilde{y}}_{i}^{p}<\underline{y_{i}} \end{array}\right., \end{array}$$
where $\bar{x_{i}}$ and $\underline{x_{i}}$ represent the lower and upper bounds of the *i*-th state, respectively, and $\bar{y_{i}}$ and $\underline{y_{i}}$ represent the lower and upper limits of the *i*-th output, respectively.

Using the metrics described above, ref. [22] proposed the following objective function to be minimized. This objective function is composed of the weighted sum of the three metrics presented:(11)$$\begin{array}{c} f\left(\Xi \right)=\underset{\Xi }{\mathrm{Min}}\left(k_{o}f_{o}\left(\cdot \right)+k_{\hat{\delta }}f_{\hat{\delta }}\left(\cdot \right)+k_{\Theta }\Theta \left(\cdot \right)\right), \end{array}$$
where Ξ are the parameters that encode the signal, and $k_{o}$, $k_{\hat{\delta }}$, and $k_{\Theta }\in R_{\geq 0}$ are constant weightings related to metrics and established according to the priority.

## 4. Meta-Heuristic Application

This section presents fundamental aspects for the application of meta-heuristics in the rSOESGOPE concept. It is divided into two parts. Section 4.1 and Section 4.2 present the signal representation and the adjustment criteria for meta-heuristics, respectively.

### 4.1. APRBS Representation and Settings

A fundamental step to applying any meta-heuristic in an OSD problem is the signal definition and its representation. These definitions determine the composition of the individuals in the technique and also the search space dimension.

In the rSOESGOPE method, the chosen signal is APRBS, which allows the incorporation of operational or safety restrictions together with the excitation persistence optimization [7,21,33]. It is a fact that, in principle, unplanned signals cannot be guaranteed.

The APRBS strategy generates a coding vector $\Xi_{k}\in R^{2\times k}$, defined by *k* stages or pairs of amplitude and time interval, mathematically represented by [21]
(12)$$\Xi_{k}=\left[t_{1},t_{2},\ldots ,t_{k},A_{1},A_{2},\ldots ,A_{k}\right],$$
where $\left[t_{1},t_{2},\ldots ,t_{k}\right]$ are the interval times and $\left[A_{1},A_{2},\ldots ,A_{k}\right]$ are the respective amplitudes.

Therefore, this APRBS signal with six stages yields an encoded signal $\Xi_{6}\in R^{12}$ or $\Xi_{6}=\left[A_{1},\ldots ,A_{6},t_{1},\ldots ,t_{6}\right]$. Figure 2 shows this example, where the realization of these parameters in $u(\Xi_{6})$ is marked as black.

Thus, the proposed challenge for the meta-heuristics is the determination of the elements $\Xi_{k}$, whose dimension depends on the number of stages *k*, where defining the number of stages is not a trivial task. As the object of study can vary, it is difficult to determine or establish a general rule.

Furthermore, it is known that the greater the number of signal stages, the greater the chance of producing signals with a wide range of amplitude and frequency. However, the larger the solution space is, the greater the effort required by the meta-heuristic, and therefore, the greater the number of individuals in the population should be.

To address this situation, it is advisable to conduct systematic tests that analyze different configurations of the stages and population numbers, as well as the study of the necessary signals for $U^{⊕}$. Therefore, by seeking a feasible trade-off relationship, this procedure ensures more accurate and appropriate definitions for the problem.

### 4.2. Meta-Heuristic Parameters

To ensure a fair comparison between the techniques, the algorithms were implemented with the same number of iterations and population size.

By default, the number of iterations is set to 100, while the population size was determined based on the study presented in Section 5.2. This study aimed to identify the optimal configuration, especially regarding the number of signal stages, which significantly impacts the solution space.

The remaining control parameters were defined in accordance with the original algorithm guidelines. You can find these parameters listed in Table 1.

## 5. Real Case Study: AERO4River

This section presents a study of metaheuristics applied in the rSOESGOPE concept. A real case study of the AERO4River ASV modeling is presented and discussed.

For this purpose, Section 5.1 introduces the AERO4River ASV and its mathematical model. Section 5.2 describes the procedures for applying the rSOESGOPE method and generating robust signals. Section 5.3 presents the results and implications of using multiple signals for each investigated optimization technique. Section 5.4 compares the results of each obtained model $M({\hat{\Gamma }}^{+})$ with the real ASV, i.e., R(Γ) for different input signals.

### 5.1. AERO4River ASV

The AERO4River ASV is a vessel developed to perform autonomous hydrological measurements in environments with underwater obstacles and shallow and rapid water flows. It is based on a catamaran-type vessel and features an innovative air propulsion system with azimuth control, resulting in a highly maneuverable ASV with three over-actuated DoFs, capable of meeting the required operating conditions. Figure 3 illustrates it.

For more constructive information, it is recommended to consult the research in [22,34,35].

#### Kinematics and Dynamics Modeling

In the context of surface vessels, modeling requires the development of a description with three Degrees of Freedom (DoFs), namely, translation movements along the *x* and *y* axes ( surge and sway) and rotation around the *z*-axis ( yaw). Other Degrees of Freedom (DoFs) are disregarded due to their minimal influence on the vessel’s dynamics [36].

According to marine vehicle nomenclature, it is customary to express state vectors as follows: $\eta ={\left[x,y,\psi \right]}^{T}$, representing inertial positions (x,y) and angular position (ψ) in the vehicle’s inertial frame; $\nu ={\left[u,v,r\right]}^{T}$, representing linear velocities (u,v) and angular velocity (r) in the rigid-body frame [37].

The general ASV dynamics and kinematics modeling without disturbance is represented as follows [38]: (13)$$\begin{array}{ccc} \left[M_{\mathrm{RB}}+M_{A}\right]\dot{\nu }+\left[C_{\mathrm{RB}}\left(\nu \right)+C_{A}\left(\nu \right)+D\left(\nu \right)\right]\nu & = & \tau , \end{array}$$(14)$$\begin{array}{ccc} \dot{\eta } & = & J(\psi )\nu , \end{array}$$
where $M_{\mathrm{RB}}\in R^{3\times 3}$ is the rigid-body Inertia Matrix; $C_{\mathrm{RB}}\left(\nu \right)\in R^{3\times 3}$ is the Coriolis and centripetal rigid-body Matrix; J(ψ) is the Jacobian matrix that relates the velocities in the rigid-body and inertial frame; the matrices $M_{A}\in R^{3\times 3}$ and $C_{A}\left(\nu \right)\in R^{3\times 3}$ represent the added mass phenomenon; $D(\nu )\in R^{3\times 3}$ represents the hydrodynamic damping; $\tau \in R^{3\times 1}$ is the vector of generalized forces and moments applied by the propellant system.

The matrices $M_{\mathrm{RB}}$, $C_{\mathrm{RB}}\left(\nu \right)$, and J(ψ) are represented by [38]
(15)$$\begin{array}{ccc} M_{\mathrm{RB}} & = & \left[\begin{array}{ccc} m & 0 & 0\\ 0 & m & mx_{g}\\ 0 & mx_{g} & I_{z} \end{array}\right], \end{array}$$
(16)$$\begin{array}{ccc} C_{\mathrm{RB}}\left(\nu \right) & = & \left[\begin{array}{ccc} 0 & 0 & -m(x_{g}r+v)\\ 0 & 0 & mu\\ m(x_{g}r+v) & -mu & 0 \end{array}\right], \end{array}$$
(17)$$\begin{array}{ccc} J(\psi ) & = & \left[\begin{array}{ccc} \cos(\psi ) & -\sin(\psi ) & 0\\ \sin(\psi ) & \cos(\psi ) & 0\\ 0 & 0 & 1 \end{array}\right], \end{array}$$
where *m* is the vehicle total mass, $I_{z}$ is the *z*-axis moment of inertia, and $x_{g}$ is the gravity center displacements on the *x*-axis.

As for the $M_{A}$ and $C_{A}\left(\nu \right)$ matrices (which are related to the added mass phenomenon), they are defined by [38]
(18)$$\begin{array}{ccc} M_{A} & = & -\left[\begin{array}{ccc} X_{\dot{u}} & {\bar{X}}_{\dot{v}} & {\bar{X}}_{\dot{r}}\\ {\bar{Y}}_{\dot{u}} & Y_{\dot{v}} & {\bar{Y}}_{\dot{r}}\\ {\bar{N}}_{\dot{u}} & {\bar{N}}_{\dot{v}} & N_{\dot{r}} \end{array}\right], \end{array}$$
(19)$$\begin{array}{ccc} C_{A}\left(\nu \right) & = & \left[\begin{array}{ccc} 0 & 0 & -\alpha_{2}\\ 0 & 0 & \alpha_{1}\\ \alpha_{2} & -\alpha_{1} & 0 \end{array}\right], \end{array}$$
where $X_{\dot{u}},Y_{\dot{v}},N_{\dot{r}}$ represent the decoupling parameters and the symmetry of the vessel, while the other parameters express the vessel’s asymmetries.

Each element of $M_{A}$ represents the mass added in a specific direction. For instance, $Y_{\dot{r}}$ corresponds to the added mass affecting the *y*-axis due to the acceleration $\dot{r}$. Furthermore, it is defined as $\alpha_{1}=X_{\dot{u}}u+X_{\dot{v}}v+X_{\dot{r}}r$ and $\alpha_{2}=Y_{\dot{u}}u+Y_{\dot{v}}v+Y_{\dot{r}}r$.

Finally, the hydrodynamic damping D(ν) is expressed as a combination of linear and nonlinear damping, which is directly related to quadratic damping. Thus, D(ν) is given by [38]
(20)$$\begin{array}{c} \begin{array}{c} D(\nu )=\left[\begin{array}{ccc} -X_{u} & -{\bar{X}}_{v} & -{\bar{X}}_{r}\\ -{\bar{Y}}_{u} & -Y_{v} & -{\bar{Y}}_{r}\\ -{\bar{N}}_{u} & -{\bar{N}}_{v} & -{\bar{N}}_{r} \end{array}\right]+\left[\begin{array}{ccc} -X_{|u|u}\left|u\right| & -{\bar{X}}_{|u|v}\left|u\right| & -{\bar{X}}_{|u|r}\left|u\right|\\ -{\bar{Y}}_{|v|u}\left|v\right| & -Y_{|v|v}\left|v\right| & -{\bar{Y}}_{|v|r}\left|v\right|\\ -{\bar{N}}_{|r|u}\left|r\right| & -{\bar{N}}_{|r|v}\left|r\right| & -N_{|r|r}\left|r\right| \end{array}\right], \end{array} \end{array}$$
where the elements of linear damping $X_{u}$, $Y_{v}$, and $N_{r}$ represent the uncoupled linear parameters and the vessel’s symmetry, while ${\bar{X}}_{v}$, ${\bar{X}}_{r}$, ${\bar{Y}}_{u}$, ${\bar{Y}}_{r}$, ${\bar{N}}_{u}$, and ${\bar{N}}_{v}$ represent its asymmetries.

For large-scale boats, the asymmetries can be disregarded in the formulation, simplifying the linear damping relationship for a given direction of movement. Additionally, in the nonlinear damping, $X_{|u|u}$, $Y_{|v|v}$, and $N_{|r|r}$ represent the nonlinear symmetrical and uncoupling damping of quadratic order in the directions of the *x*, *y*, and *z* axes, respectively. The remaining parameters represent the asymmetries and couplings.

Assuming that $M_{\mathrm{RB}}$ and $C_{\mathrm{RB}}\left(\nu \right)$ are related to previously known variables, the parameter identification problem boils down to finding $M_{A},C_{A}\left(\nu \right)$, and D(ν). Therefore, the solution to the parameter identification problem exists within a $R^{27}$ dimensional space or finding of the hydrodynamic derivatives.

### 5.2. rSOESGOPE Application

To apply the rSOESGOPE method, the first step is to obtain an initial estimate of the vessel’s dynamics. Based on this estimation, benchmark parameters are designed for later use in designing identification signals, applying them to the vessel, and ultimately estimating the posteriori parameters.

Thus, the application of the method follows a flowchart with four well-defined steps: (1) Prior Estimation, (2) Generate Benchmark Parameters, (3) Signal Design, and (4) Parametric Estimation. These steps are also illustrated in Figure 4.

#### 5.2.1. Priori Estimation

Two approaches were employed to obtain the initial estimation of the vessel’s dynamics.

First, the operational and inertial parameters were obtained through direct measurements or Computational Fluid Dynamics (CFD) simulations. The measured parameters are presented in Table 2, while the computationally simulated ones can be found in Table 3. In these tables, $\left[F_{x},F_{y},\tau_{n}\right]$ represent the propulsion system capacity for surge, sway, and yaw, respectively. Additionally, $\left[t_{F_{x}},t_{F_{y}},t_{\tau_{n}}\right]$ denote the time constants related to the dynamics of surge, sway, and yaw, respectively.

Subsequently, to obtain the initial values of the main hydrodynamic parameters for the simpler uncoupled model with nine variables, an experimental test was conducted using an APRBS-based signal [39]. The results of this test are presented in Table 4.

#### 5.2.2. Generation of Benchmark Parameters

To define $P^{⊕}=\left[{{\tilde{\Gamma }}^{p}}_{1},{{\tilde{\Gamma }}^{p}}_{2},\ldots ,{{\tilde{\Gamma }}^{p}}_{n}\right]$, it is first necessary to determine the number of characters *n* that make up $U^{⊕}$. This number should be sufficient to minimize uncertainties about ${\hat{\Gamma }}^{-}$ and ensure the subsequent robustness of the method.

For the presented case study, the use of up to five signals in the composition of $U^{⊕}$ will be investigated. Therefore, $P^{⊕}$ is also composed of five parameter sets ${\tilde{\Gamma }}^{p}$, which were generated by five random samples belonging to the uncertainty region of 50% around ${\hat{\Gamma }}^{-}$. The equation used is shown below:(21)$${\tilde{\Gamma }}^{p}=\left(1+\frac{\sigma }{100}r\right){\hat{\Gamma }}^{-},$$
where σ∈R represents the level of parametric uncertainty in the range of [0,100], and $r\in R^{9\times 1}$ is a vector of random numbers with uniform distributions ([−1,1]).

Table 5 shows the generated and required $P^{⊕}$ for the design of $U^{⊕}$ according to Section 3.

#### 5.2.3. Designing Signals

Firstly, it is necessary to determine the number of stages in the signal and the metaheuristic’s population size.

As shown in Section 4, the number of stages affects the elements of the individuals, i.e., the dimension of Ξ, and also the excitation’s persistence property. Moreover, the number of individuals also affects the metaheuristic’s ability to search for signals with high excitation persistence.

Therefore, in order to find the best configuration, a study was developed using the PSO, which includes a single-signal design. For a better signal persistence analysis and its relationship with the number of stages and the swarm size, no parametric constraints are considered. With this in mind, the following definitions were used:PARTICLE SWARM OPTIMIZATION (PSO) ALGORITHM: Particle population analysis ($N_{pop}$) from 5 to 20, coding each particle $\Xi_{i}$ with stages from 3 to 24 ($\Xi_{3},\Xi_{6},\Xi_{12},\Xi_{24}$), with a stop criterion of 100 generations.INTERIOR POINT ALGORITHM: $f_{o}<10^{-2}$ and μ<0.001 (μ: barrier parameter used with the convergence criterion in the Interior Points method) as stopping criteria.OBJECTIVE FUNCTION WEIGHTINGS: $k_{o}=1$, $k_{\hat{\delta }}=20$, $k_{\Theta }=0$.ERROR WEIGHTINGS: Only the velocities in the rigid-body frame [u,v,r] were considered for the optimization process, weighted by the inverse of the variance. Therefore, $Q_{x}$ was disregarded, and $Q_{y}$ was defined as follows:
(22)$$Q_{y}=\mathrm{diag}\left(\frac{1}{{\hat{\sigma }}_{u}^{2}},\frac{1}{{\hat{\sigma }}_{v}^{2}},\frac{1}{{\hat{\sigma }}_{r}^{2}}\right),$$
where ${\hat{\sigma }}_{u}=0.05$ [m/s], ${\hat{\sigma }}_{v}=0.05$ [m/s], and ${\hat{\sigma }}_{r}=0.01$ [rad/s] are estimated standard deviations of the measurement errors of surge, sway, and angular rate, respectively.SIGNAL CODING: The OSD problem consists of the determination of $u(\Xi_{k})$, where $\Xi_{k}$ is defined as follows:
(23)$$\Xi_{k}=\left[\begin{array}{cccccc} t_{Fx}^{1}, & \ldots , & t_{Fx}^{k}, & F_{x}^{1}, & \ldots , & F_{x}^{k}\\ t_{Fy}^{1}, & \ldots , & t_{Fy}^{k}, & F_{y}^{1}, & \ldots , & F_{y}^{k}\\ t_{\tau_{n}}^{1}, & \ldots , & t_{\tau_{n}}^{k}, & \tau_{n}^{1}, & \ldots , & \tau_{n}^{k} \end{array}\right],$$
where *k* is the stage number, $\left[F_{x}^{k},F_{y}^{k},\tau_{n}^{k}\right]$ are the APRBS amplitudes, and $\left[t_{F_{x}}^{k},t_{F_{y}}^{k},t_{\tau_{n}}^{k}\right]$ are the APRBS swap times.

Table 6 presents the simulation results for 50 iterations (referred to as ‘50 signals’) for each configuration under study. These results are represented by the average (av) and standard deviation (std) of f(Ξ). Additionally, all simulations utilized the same sets of initial parameters ${\hat{\Gamma }}^{-}$ (refer to Table 4) and reference ${\tilde{\Gamma }}^{p}$ (as shown in Equation (Equation 21)). Furthermore, they employed identical initial conditions for generating the initial signals.

Table 6 demonstrates an improvement in signal persistence as the number of particles increases. This improvement is noticeable from 5 to 20 particles, with significant enhancements. Additionally, it is observed that there is an enhancement in signal quality up to six stages. Beyond this threshold, the signal’s persistence decreases, indicating that further increasing the signal encoding is no longer beneficial.

Consequently, following the proposed framework, the optimal configuration for this problem involves using 20 particles and six stages.

Based on these findings, five signals were generated for each metaheuristic. Essentially, each signal ${u^{⊕}}_{i}$ from $U^{⊕}=\left[{u^{⊕}}_{1},{u^{⊕}}_{2},\ldots ,{u^{⊕}}_{5}\right]$ is individually designed for a benchmark parameter ${{\tilde{\Gamma }}^{p}}_{i}$ from $P^{⊕}=\left[{{\tilde{\Gamma }}^{p}}_{1},{{\tilde{\Gamma }}^{p}}_{2},\ldots ,{{\tilde{\Gamma }}^{p}}_{5}\right]$. Therefore, Equation (Equation 5) can also be applied without loss of generality as follows:(24)$$\begin{array}{c} {u^{⊕}}_{i}=S(M\left({\hat{\Gamma }}^{-}\right),M\left({{\tilde{\Gamma }}^{p}}_{i}\right)), \end{array}$$
where S represents the double-layer optimization methodology (referred to as “Metaheuristic + IPA”) whose metaheuristics will be investigated. Additionally, parametric constraints have been applied to generate achievable signals that closely resemble the ASV behavior, resulting in $k_{\theta }=2\times 10^{5}$.

Figure 5, Figure 6, Figure 7, Figure 8, Figure 9, Figure 10, Figure 11 and Figure 12 display the generated APRBS signals. These figures exhibit the typical characteristics of projected signals, including multiple amplitudes and varying excitation frequencies. It is worth noting that, for the analyzed vessel, the total experiment time does not exceed 32 s. This observation is noteworthy because there is often a natural assumption that longer excitation signals are superior to shorter ones. However, the results suggest that this is not an absolute truth. Experiments with excessively long duration may not necessarily enhance parameter learning but rather contribute to the richness of excitation in the projected signals. Achieving this richness is possible through the optimization process, even in shorter experiments, as long as the quality of excitation is maintained.

Table 7 presents the properties of the projected signals across all studied metaheuristics.

Table 7 presents the performance of the generated signals for each ${{\tilde{\Gamma }}^{p}}_{i}$. It is also evident that the reference parameters used impose varying degrees of difficulty. Notably, the ${{\tilde{\Gamma }}^{p}}_{2}$ sample posed the greatest challenges for the methods, particularly for the GA, which struggled to satisfy the boundary conditions. Additionally, the last column of the table displays the total sum of f(Ξ) for all five signals. In this regard, it can be observed that the GA and AOA techniques yielded the poorest results. Conversely, GWO, PSO, BA, and MTDE produced very similar results, with PSO achieving the best outcome.

### 5.3. Parametric Estimation

After designing $U^{⊕}$, the final step is to apply it to the real ASV, denoted as R(Γ), and perform the PE. In this case, the PE problem aims to determine $\Gamma \in R^{27}$ by exploring the capabilities of the rSOESGOPE method, as specified in Section 5.2. It is possible to analyze the PE results using up to five optimal signals. Therefore, for each ${U^{⊕}}_{n}$, a posteriori estimate ${\hat{\Gamma }}^{+}$ is computed from
(25)$$\begin{array}{ccc} {\hat{\Gamma }}_{n}^{+} & = & P(M\left({\hat{\Gamma }}^{-},{U^{⊕}}_{n}\right),R\left(\Gamma ,{U^{⊕}}_{n}\right)), \end{array}$$
where n represents the number of signals of $U_{n}^{⊕}=\left[{u^{⊕}}_{1},\ldots ,{u^{⊕}}_{n}\right]$ and P represents the IPA responsible for solving the PE problem, as shown in Section 3.

For comparison purposes, the ${\hat{\Gamma }}^{+}$ obtained by each method is evaluated against the scenario provided by $R(\Gamma ,U^{⊕})$. The results are presented in Table 8, where the Root-Mean Square Error (RMSE) measures the performance.

The results in Table 8 reveal that using more than one signal in $u^{⊕}$ led to improvements in the final estimate of ${\hat{\Gamma }}^{+}$. This improvement was observed for the GWO, WOA, PSO, SSA, BA, and MTDE methods, where three or more signals were used, resulting in enhancements. However, the same trend was not observed for the AOA and GA techniques, as the addition of signals worsened their performance with each addition to ${U^{⊕}}_{n}$. This phenomenon can be attributed to the difficulties these techniques face in the design phase, as indicated in Table 7.

Based on these results, it is also evident that the use of multiple signals led to improvements in six out of the eight metaheuristics. It is important to note that the addition of signals should be carried out cautiously and in moderation, as seen in the case of WOA, where using more than four signals was no longer beneficial. These findings suggest that adding signals without caution can either amplify already identified features or introduce new dynamics to the estimated model.

Table 9 and Table 10 present the best estimates obtained based on the results from Table 8. Consequently, the GWO technique employs five signals, Whale Optimization Algorithm (WOA) uses three signals, and PSO employs five signals. The AOA and GA techniques utilize only one signal, while SSA, BA, and MTDE utilize four signals.

### 5.4. Model Validation

To evaluate the quality of the models obtained from each meta-heuristic, three different maneuver signals were employed (All experimental data, including the ASV’s states and signals, are available in the dataset [40]). In scenario 1 (C1), the ASV undergoes frontal motion with small yaw corrections. In scenario 2 (C2), the vessel experiences the traditional zigzag motion. Finally, in the last scenario (C3), the ASV is subjected to coupled motions.

Table 11 shows the models’ performance, while the signals used and the behavior of the estimated models can be seen in Figure 13 and Figure 14. In addition, the performance of each model $M({\hat{\Gamma }}^{+})$ is measured by RMSE.

Analyzing the experimental results and their performance in Table 11, it can be concluded that the majority of metaheuristics yielded good results. The exceptions are the AOA and BA methods, as evident in Figure 14. In this figure, it is apparent that the AOA failed to capture the primary dynamics of the ASV, resulting in a significant divergence in 3 DoFs. On the other hand, the Bat Algorithm (BA) method demonstrated strong generalization abilities for sway and yaw motions but exhibited a high degree of divergence in the surge DoF during Track 3.

As for the other techniques, it is noteworthy that the estimated models exhibited behaviors closely resembling the ASV in 3 DoFs, as illustrated in Figure 13 and Figure 14. Among these techniques, it is worth mentioning that the GWO and MTDE methods delivered the best performances and outperformed the PSO. According to Table 11, GWO reduced the RMSE by approximately 6.21%, while MTDE reduced it by 0.69% when compared to PSO.

Another fact showing that the results obtained were satisfactory is found in Table 12, which shows the RMSE of each DoF over all experiments.

Considering that the experiments were conducted in an open environment exposed to wind, waves, and other external disturbances, the results remain consistent, except for the AOA and BA techniques. For instance, in the case of GWO, an RMSE of 0.24 [m/s] was obtained for surge, and 0.20 [m/s] for sway, which correspond to errors of 12% and 10%, respectively, when considering a linear speed operation of 2.00 [m/s]. Regarding yaw movement, the RMSE error of 0.34 [rad/s] would equate to 11.85% if the ASV operated at full capacity. Hence, both results are acceptable under the presented conditions, and the same holds true for methods using multiple signals, as demonstrated in Figure 13 and Figure 14.

From Table 12, it can be observed that the methods using multiple signals (GWO, WOA, PSO, SSA, BA, and Multi-trial vector-based Differential Evolution (MTDE)) yield better results compared to the estimation models using a single signal (AOA and GA). For example, when comparing the results of GA with those of GWO, a reduction in RMSE of 14.28% is observed in the surge DoF, and similar improvements are observed in other DoFs.

## 6. Qualitative Analysis of Algorithm Performance

Taking a closer look at the algorithmic results, it is worthwhile to compare the techniques that achieved success, referred to as Group 1 (GWO, WOA, PSO, SSA, BA, and MTDE) and Group 2 (AOA and GA), compared to techniques that did not perform well in the studied problem.

In the case of Group 2, the GA exhibited low performance. Despite its wide applicability in complex problems, it is known that the GA can face challenges related to parameterization and slow convergence, especially in complex and multidimensional problems. These challenges became evident in the final model of the algorithm, which was developed using only one optimal excitation signal. Given the complexity of the problem, this resulted in a model with limited overall learning and reduced performance.

Similarly, within Group 2, the AOA also failed to achieve satisfactory results. Premature convergence and difficulties in exploring the search space were observed. These issues can be attributed to the use of only one signal in constructing the model. Similar to the GA, this led to a model with limited overall learning. Contributing factors include the dependence on the exploration–exploitation trade-off of two parameters with fixed values, the absence of information sharing and guidance towards the best global solution, and the modification of only the best solution based on arithmetic operators during the exploration and exploitation stages of the algorithm.

Turning to the results from Group 1, it is evident that all meta-heuristics generated models using three or more sub-optimal signals. This indicates that these algorithms did not encounter issues with premature convergence or difficulties in exploring the solution space. Furthermore, they allowed for the development of models with increased learning through the addition of new signals.

Among the six techniques in Group 1, three stood out as highly efficient: GWO, and the meta-heuristics MTDE and PSO. These techniques yielded very similar results and demonstrated efficiency in exploring the search space and addressing the complexities of OSD problems. GWO achieved the best performance, which can be attributed to (i) its lower sensitivity to parameters, making it easier to configure and fine-tune compared to other algorithms, and (ii) its ability to explore a wide range of solutions within its search space before converging towards solutions closer to the global optimum.

It is important to emphasize that each algorithm possesses individual characteristics and may excel in different scenarios. In this study, GWO demonstrated superiority, even outperforming the originally used PSO in the rSOESGOPE method. Finally, for a better understanding, Table 13 below provides a comprehensive comparison of the entire study conducted in this article.

## 7. Conclusions

This paper presented a comparison of metaheuristic performance when applied to Optimal Signal Design for estimating parameters of a nonlinear system. In particular, various metaheuristics were investigated and applied in the context of rSOESGOPE, which originally utilized the PSO algorithm as the primary optimizer. To assess the performance of these metaheuristics, a modeling problem of a catamaran-type ASV was employed. From the results, several key points can be emphasized:ASV was chosen because it presents several difficulties in the scope of modeling and identification, which must deal with geometric and mechanical asymmetries, coupling between the DoFs, a high level of uncertainty about the initial estimates, and other aspects such as environmental disturbances.Furthermore, these challenges also include the design of signals in large search spaces and the parametric estimation of many parameters.The techniques were analyzed at each stage of the rSOESGOPE methodology, from configuration, signal design, and parameter estimation to the model validation obtained in different identification scenarios.Considering the Optimal Signal Design phase, it has been proposed and demonstrated that conducting the study to define the characteristics of the APRBS signal plays a fundamental role in the projected signal quality. This study shows the importance of defining the trade-off between the persistence of the excitation and the problem solvability. Although long multi-stage APRBS signals provide richer excitation characteristics, the nonlinear solution space grows exponentially, which can make it difficult, costly, or even intractable.In general, the performance of the estimated model improved with the insertion of signals, minimizing the uncertainties of the initial estimates, as initially proposed. Exceptions occurred only for the GA and AOA models, reaffirming the difficulties in the design stage regarding compliance with the established constraint restrictions and improvement of the persistence of excitation. There is a high similarity between the estimated models and the real system. In these scenarios, the GWO method presents the best RMSE among all the methods used, followed by the MTDE and PSO methods, respectively. It can also be observed that the AOA method could not find a solution that satisfies the system, as its results were about 5 times larger than the other ones. Furthermore, regarding the RMSE for each DoF, the GWO obtained the best results considering surge and sway motions and is surpassed in the yaw behavior only by the SSA method, which is still ranked only in fifth place among the metaheuristics’ results.

Finally, it is possible to appreciate the validity of the rSOESGOPE concept, which has proven its effectiveness with various algorithms. Regardless of the adopted metaheuristic, it was possible to mitigate the impact of initial uncertainties on the system and signal design, leading to the estimation of robust models. This achievement opens the door to the development of robust control strategies and fault-tolerant systems for the use of ASV in more challenging and hostile environments.

As part of future work and research, the aim is to conduct a more comprehensive assessment of the algorithms’ performance under various environmental conditions and in harsher scenarios (e.g., fast-flowing water, waves, and/or winds) to gain a more comprehensive understanding of their applicability and limitations.

Furthermore, a future project involves the creation of a hybrid metaheuristic model that combines the strengths of the top-performing algorithms. This could lead to a more robust and efficient solution, further enhancing the capabilities of our models and their adaptability to a broader range of challenges.

A schematic video demonstrating the application of the technique is available at https://www.youtube.com/watch?v=OG0gr67YMNg, accessed on 11 October 2023.

## Figures and Tables

**Figure 1 sensors-23-09085-f001:**
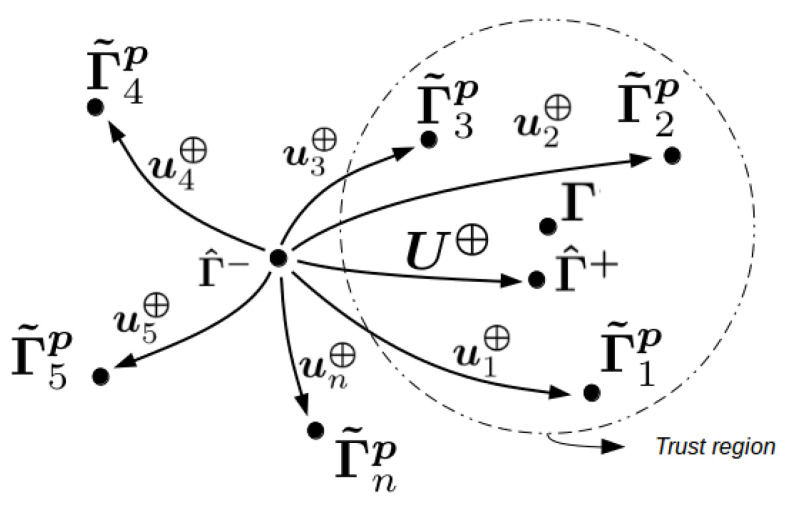
rSOESGOPE concept.

**Figure 2 sensors-23-09085-f002:**
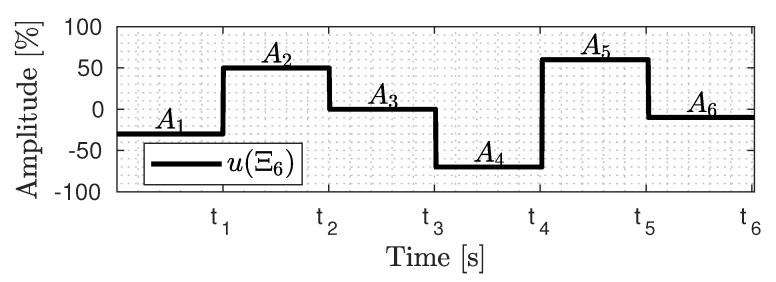
APRBS signal parameterization example.

**Figure 3 sensors-23-09085-f003:**
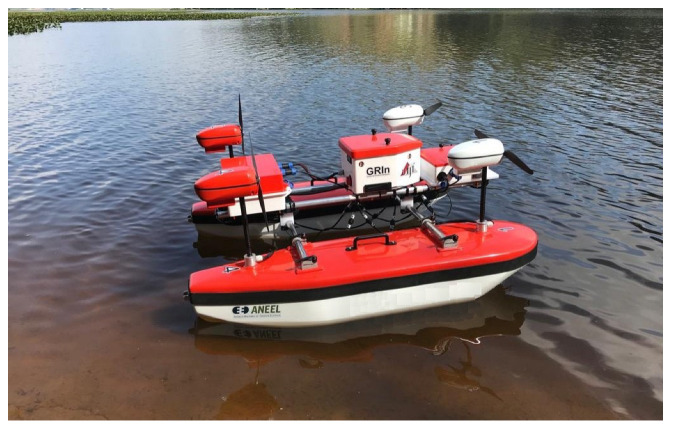
Real view of the developed catamaran.

**Figure 4 sensors-23-09085-f004:**
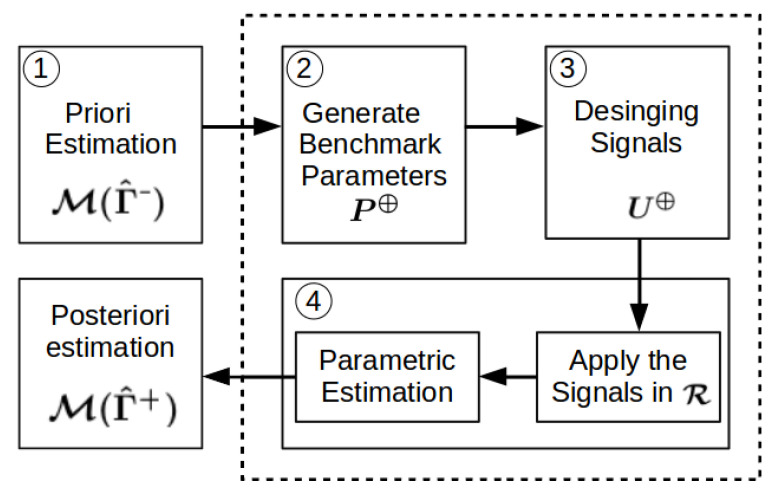
rSOESGOPE application diagram.

**Figure 5 sensors-23-09085-f005:**
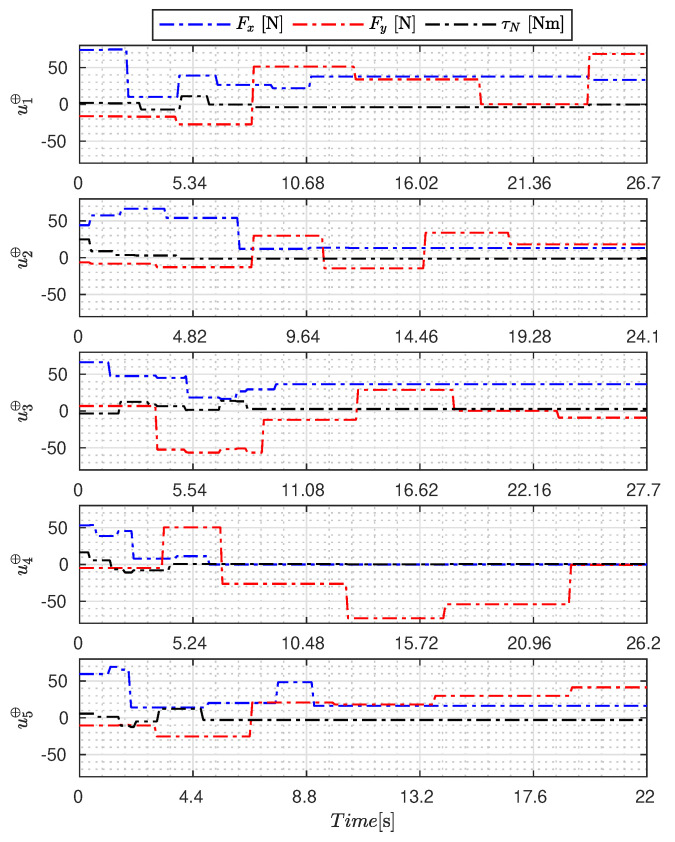
APRBS signal designed by PSO.

**Figure 6 sensors-23-09085-f006:**
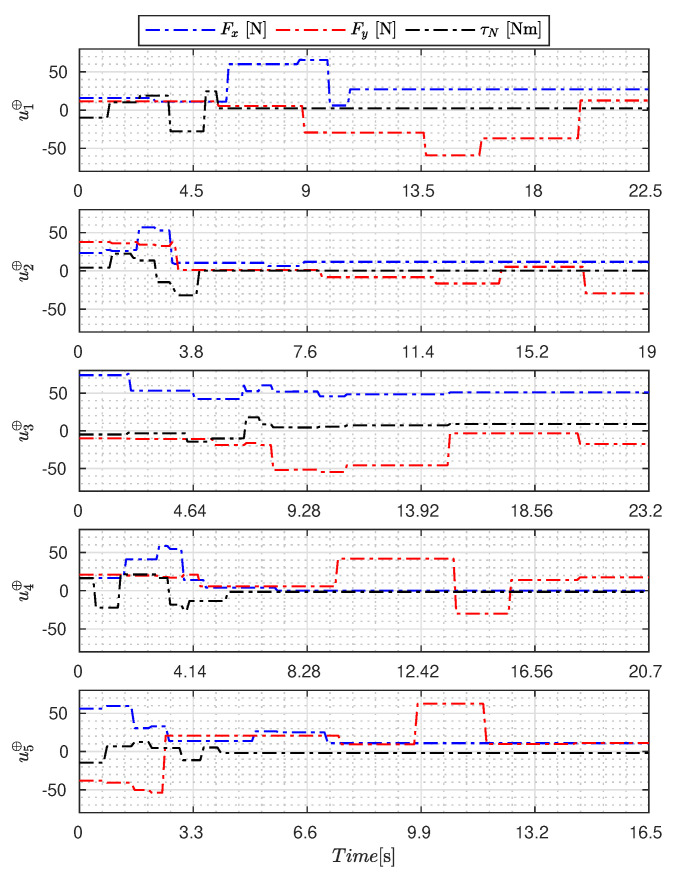
APRBS signal designed by GWO.

**Figure 7 sensors-23-09085-f007:**
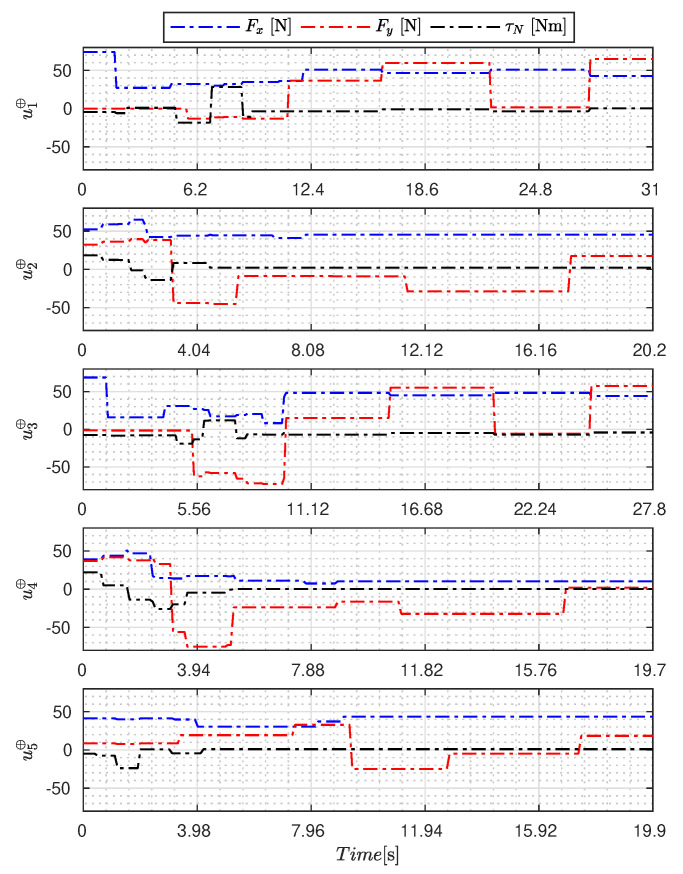
APRBS signal designed by WOA.

**Figure 8 sensors-23-09085-f008:**
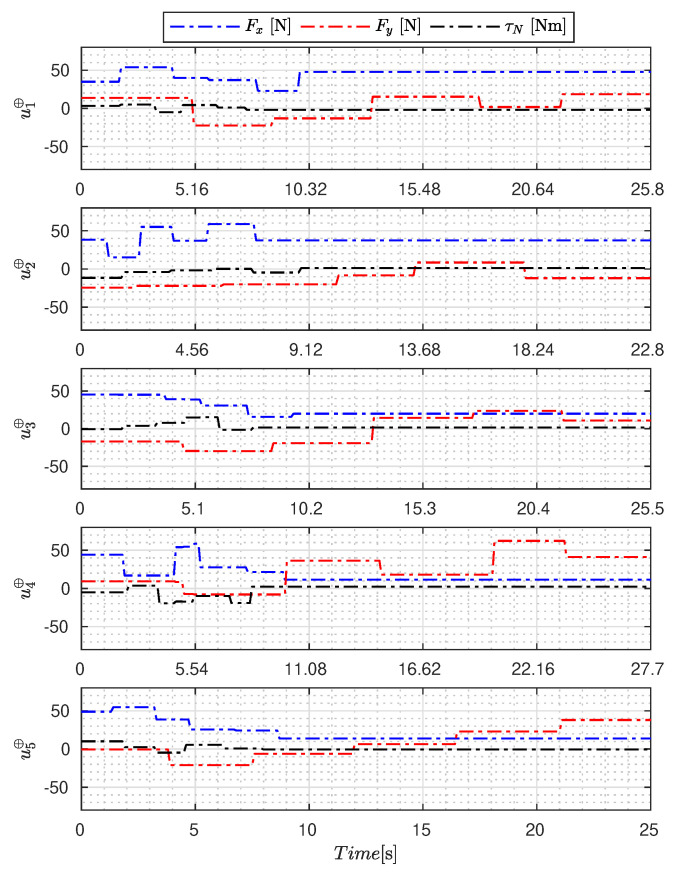
APRBS signal designed by SSA.

**Figure 9 sensors-23-09085-f009:**
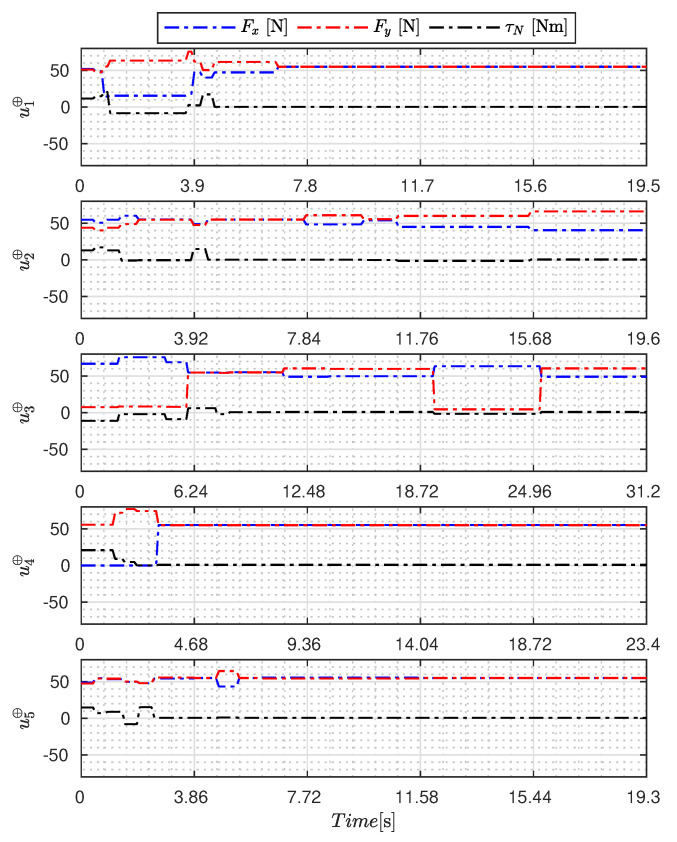
APRBS signal designed by AOA.

**Figure 10 sensors-23-09085-f010:**
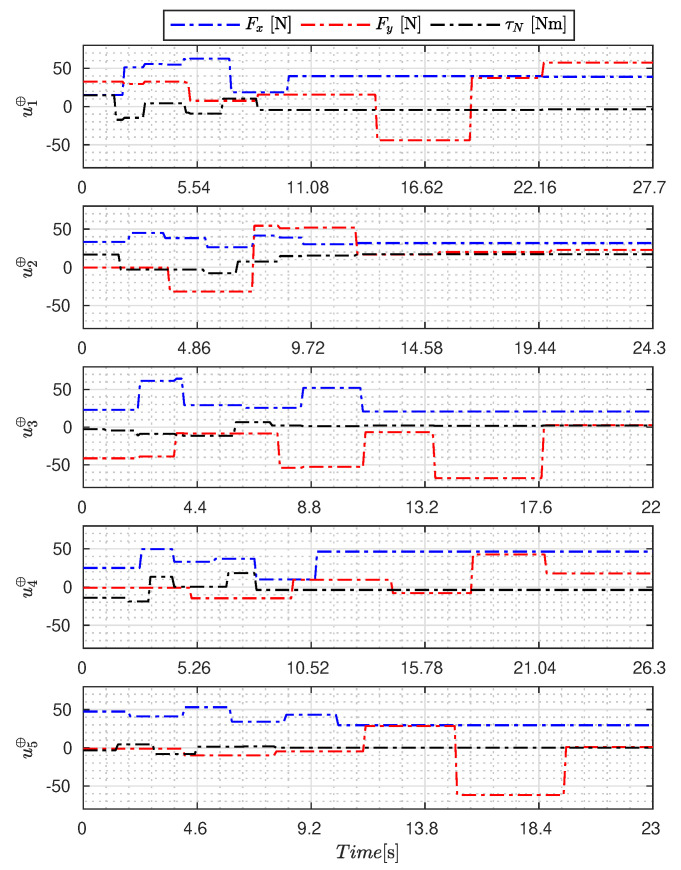
APRBS signal designed by GA.

**Figure 11 sensors-23-09085-f011:**
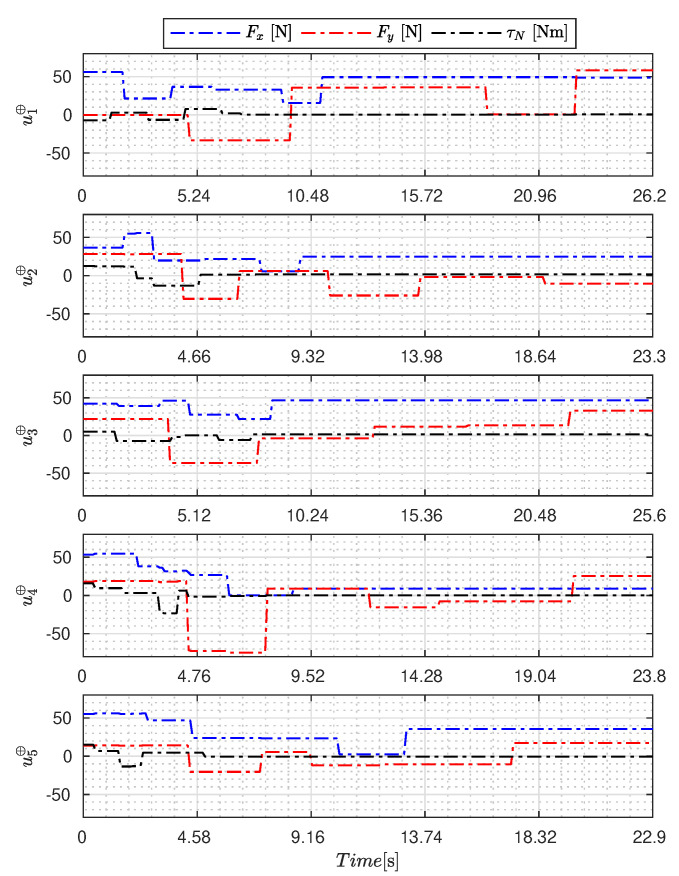
APRBS signal designed by BA.

**Figure 12 sensors-23-09085-f012:**
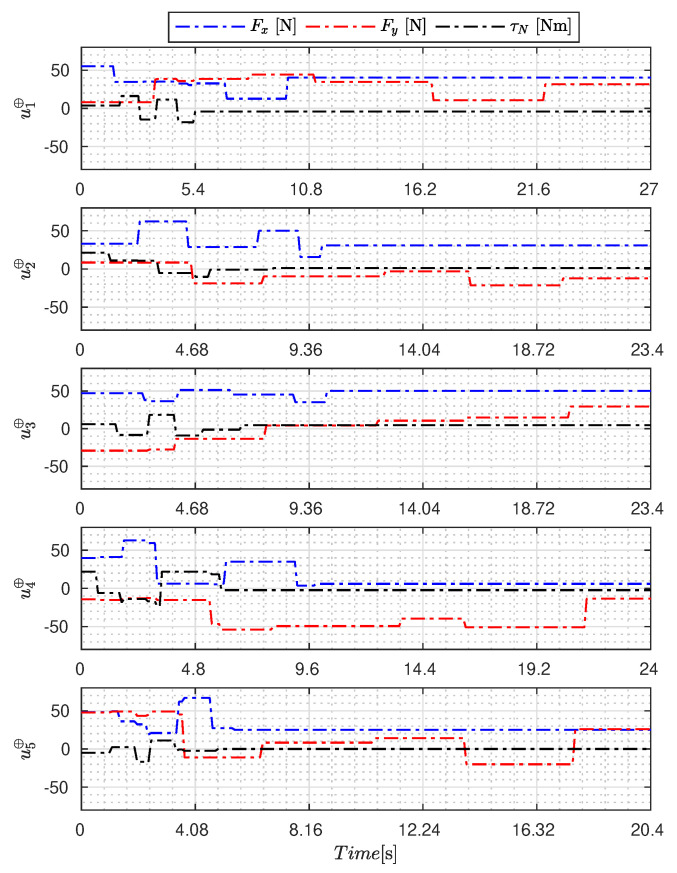
APRBS signal designed by MTDE.

**Figure 13 sensors-23-09085-f013:**
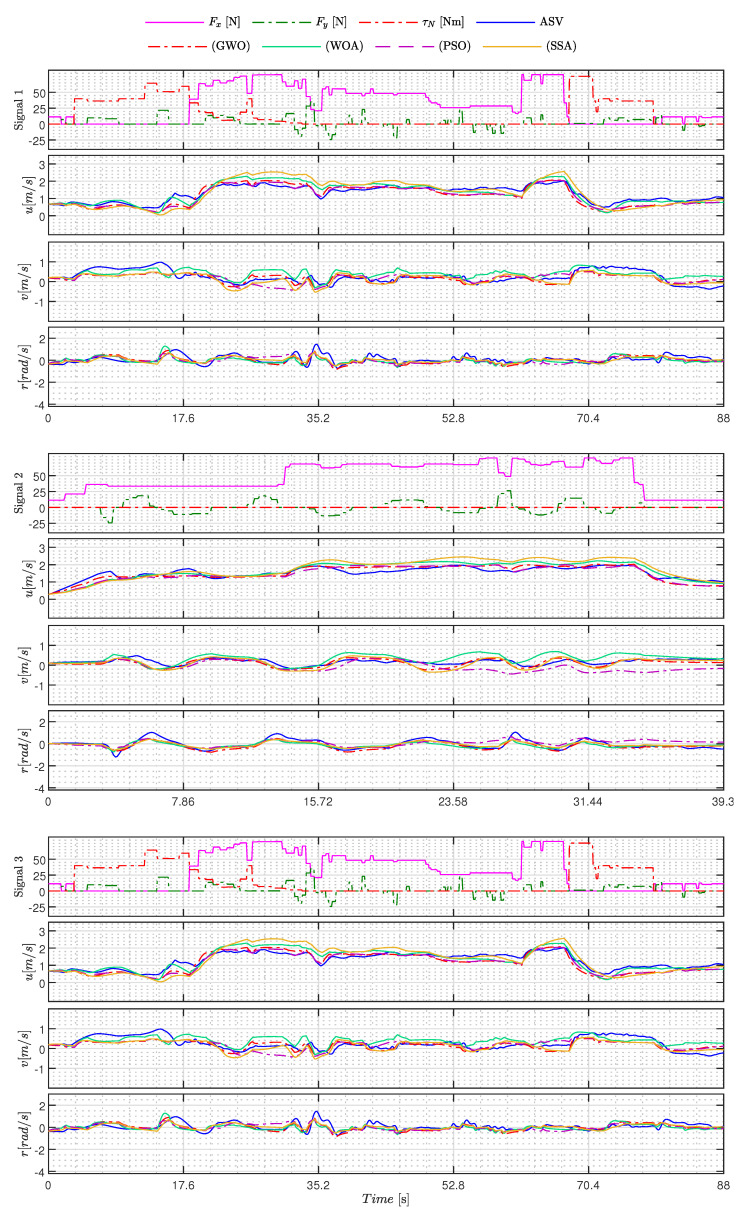
Excitationsignals and estimated states of the best GWO, WOA, PSO, and SSA models for C1 (Signal 1), C2 (Signal 2), and C3 (Signal 3).

**Figure 14 sensors-23-09085-f014:**
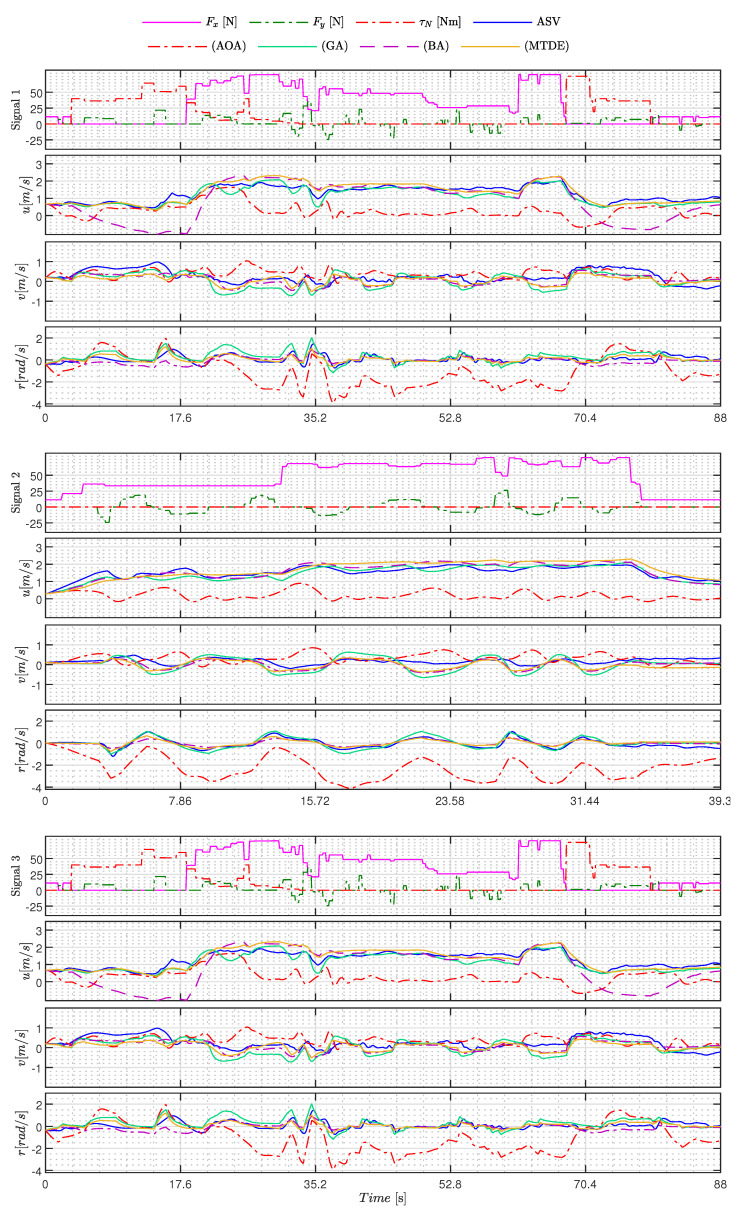
Excitationsignals and estimated states of the best AOA, GA, BA, and MTDE models for C1 (Signal 1), C2 (Signal 2), and C3 (Signal 3).

**Table 1 sensors-23-09085-t001:** Parameter values for the comparative algorithms.

	Parameter (Characteristic)
GWO	Convergence parameter (*a*): Linear reduction from 2 to 0
WOA	Convergence parameter (*a*): Linear reduction from 2 to 0
	Convergence parameter ($a_{2}$): Linear reduction from −1 to −2
PSO	Topology: Fully connected
	Cognitive and social constant: $c_{1}=2,c_{2}=2$
	Inertia weight (Linear reduction): From 0.9 to 0.2
	Velocity limit: 20% of dimension range
SSA	Convergence parameter (*a*): nonlinear reduction from 2 to 0
AOA	Math Optimizer probability: from 0.2 to 0.9
	Exploitation accuracy parameter: α=10
	Search process parameter: μ=0.499
GA	Type: Encoding Arithmetic
	Selection: Roulette wheel
	Crossover—Whole arithmetic: Probability = 0.95
	Mutation: Probability = 0.01
	Elitism: 20% of population
BA	Frequency Range: 0 (min) to 1 (max)
	Reduction in loudness rate: α=0.9
	Increase in emission rate: λ=0.1
MTDE	Number of improvement (TVP): WinIter = 20
	Best individual number: H=5
	Initial and final value of $a_{2}$: [0.001, 2]
	Decrease coefficient of $a_{2}$: Mu (D) = log(D)

**Table 2 sensors-23-09085-t002:** Vessel inertial parameters.

Parameter	Value	Unit
*m*	33.06	[kg]
$I_{z}$	4.12	[kg·m${}^{2}$]
$x_{g}$	−0.05	[m]
$y_{g}$	0.00	[m]

**Table 3 sensors-23-09085-t003:** Vessel operating characteristics.

Variable	Inferior Limit	Upper Limit	Unit
*u*	0.00	2.05	[m/s]
*v*	−0.90	0.90	[m/s]
*r*	−2.87	2.87	[rad/s]
$F_{x}$	0.0	78.0	[N]
$F_{y}$	−78.0	78.0	[N]
$\tau_{n}$	−42.0	42.0	[N·m]
$t_{F_{x}}$	0.75	3.15	[s]
$t_{F_{y}}$	1.95	5.85	[s]
$t_{\tau_{n}}$	0.45	2.70	[s]

**Table 4 sensors-23-09085-t004:** Summary of the Prior Estimation $M({\hat{\Gamma }}^{-})$.

Parameter	Value	Unit
$X_{\dot{u}}$	−34.72	[kg]
$X_{u}$	−0.00	[kg · s${}^{-1}$]
$X_{|u|u}$	−15.51	[kg · m${}^{-1}$]
$Y_{\dot{v}}$	−33.78	[kg]
$Y_{v}$	−0.00	[kg · s${}^{-1}$]
$Y_{|v|v}$	−86.87	[kg · m${}^{-1}$]
$N_{\dot{r}}$	−8.58	[kg· m${}^{2}\cdot$ rad${}^{-1}$]
$N_{r}$	−5.44	[kg· m${}^{2}\cdot$ rad${}^{-1}\cdot$ s${}^{-1}$]
$N_{|r|r}$	−1.01	[kg · m${}^{2}\cdot$ rad${}^{-2}$]

**Table 5 sensors-23-09085-t005:** Generated benchmark parameters—$P^{⊕}$.

-	${{\tilde{\Gamma }}^{p}}_{1}$	${{\tilde{\Gamma }}^{p}}_{2}$	${{\tilde{\Gamma }}^{p}}_{3}$	${{\tilde{\Gamma }}^{p}}_{4}$	${{\tilde{\Gamma }}^{p}}_{5}$
$X_{\dot{u}}$	−43.91	−35.65	−36.58	−23.74	−25.58
$X_{u}$	−0.00	−0.00	−0.00	−0.00	−0.00
$X_{|u|u}$	−15.47	−19.05	−20.75	−13.09	−12.67
$Y_{\dot{v}}$	−42.95	−47.91	−21.63	−37.81	−25.32
$Y_{v}$	−0.00	−0.00	−0.00	−0.00	−0.00
$Y_{|v|v}$	−124.59	−59.43	−84.87	−112.75	−87.77
$N_{\dot{r}}$	−11.66	−12.58	−6.56	−10.88	−6.41
$N_{r}$	−7.89	−4.71	−7.55	−4.43	−5.13
$N_{|r|r}$	−0.85	−0.91	−1.33	−1.33	−1.26

**Table 6 sensors-23-09085-t006:** Analysis of number of stages and individuals.

$N_{\mathrm{pop}}$	$\Xi_{3}$		$\Xi_{6}$		$\Xi_{12}$		$\Xi_{24}$	
	av	std	av	std	av	std	av	std
05	0.63	0.24	0.57	0.13	2.12	0.36	1.58	0.41
10	0.48	0.15	0.47	0.10	1.24	0.43	1.26	0.31
15	0.43	0.11	0.43	0.13	0.83	0.37	1.28	0.41
20	0.40	0.10	0.39	0.11	0.64	0.20	1.12	0.36

**Table 7 sensors-23-09085-t007:** Design signal performance—f(Ξ).

-	${u^{⊕}}_{1}$	${u^{⊕}}_{2}$	${u^{⊕}}_{3}$	${u^{⊕}}_{4}$	${u^{⊕}}_{5}$	∑f(Ξ)
GWO	8.55	18.10	3.01	13.79	4.10	47.54
WOA	8.97	21.56	3.85	15.67	4.57	54.63
PSO	8.35	17.23	2.55	13.30	3.54	44.97
SSA	8.43	19.27	2.78	15.69	4.01	50.18
AOA	11.36	29.64	3.99	31.36	12.96	89.30
GA	9.67	6747.08	4.16	16.39	5.33	6782.63
BA	8.68	17.58	2.70	13.99	3.83	46.78
MTDE	8.37	18.60	2.85	13.55	3.78	47.15

**Table 8 sensors-23-09085-t008:** PE performance analysis using ${U^{⊕}}_{n}$.

-	${U^{⊕}}_{1}$	${U^{⊕}}_{1}$	${U^{⊕}}_{3}$	${U^{⊕}}_{4}$	${U^{⊕}}_{5}$
GWO	0.51	0.42	0.35	0.37	0.33
WOA	0.54	0.41	0.31	0.43	0.46
PSO	0.52	0.52	0.51	0.50	0.46
SSA	0.42	0.33	0.31	0.29	0.30
AOA	0.11	0.15	0.19	0.19	0.20
GA	0.21	0.21	0.23	0.34	0.38
BA	0.50	0.38	0.37	0.31	0.31
MTDE	0.52	0.47	0.37	0.28	0.29

**Table 9 sensors-23-09085-t009:** Parametric estimation results of the metaheuristics: GWO, WOA, PSO, and SSA.

-	${\hat{\Gamma }}_{\mathrm{GWO}}^{+}$	${\hat{\Gamma }}_{\mathrm{WOA}}^{+}$	${\hat{\Gamma }}_{\mathrm{PSO}}^{+}$	${\hat{\Gamma }}_{\mathrm{SSA}}^{+}$
$X_{\dot{u}}$	−0.00	−33.77	−27.04	−44.24
$X_{u}$	−0.00	−0.00	−0.00	−0.00
$X_{v}$	−0.01	0.00	0.00	−0.01
$X_{r}$	0.00	0.12	−0.03	0.00
$X_{|u|u}$	−18.13	−14.09	−17.33	−11.92
$Y_{\dot{v}}$	−7.74	−42.49	−48.34	−46.69
$Y_{u}$	−0.00	5.16	−0.00	1.24
$Y_{v}$	−0.00	−0.00	−0.00	−0.00
$Y_{r}$	0.34	−0.00	−0.05	−0.00
$Y_{|v|v}$	−313.31	−143.19	−287.65	−295.49
$N_{\dot{r}}$	−12.53	−3.52	−9.24	−16.38
$N_{u}$	−0.02	−0.00	0.00	−0.00
$N_{v}$	0.00	0.00	0.02	−8.02
$N_{r}$	−0.02	−0.00	−0.02	−0.01
N|r|r	−14.51	−0.13	−82.08	−20.61
${\bar{X}}_{\dot{v}}$	−0.00	−0.00	0.00	0.00
${\bar{X}}_{\dot{r}}$	0.00	0.00	−0.00	−0.00
${\bar{Y}}_{\dot{u}}$	−0.00	−0.00	−0.00	−0.00
${\bar{Y}}_{\dot{r}}$	−9.89	−22.88	−0.01	−12.45
${\bar{N}}_{\dot{u}}$	0.06	−0.02	−5.51	−0.02
${\bar{N}}_{\dot{v}}$	1.93	7.27	0.00	1.08
${\bar{X}}_{|u|v}$	−0.04	0.00	0.06	−14.34
${\bar{X}}_{|u|r}$	0.00	0.18	−7.67	0.18
${\bar{Y}}_{|v|u}$	-0.00	0.05	−0.00	0.00
${\bar{Y}}_{|v|r}$	0.00	−0.00	−0.00	−0.00
${\bar{Y}}_{|r|u}$	−5.76	−0.04	0.00	−2.23
${\bar{Y}}_{|r|v}$	0.01	3.94	50.00	0.02

**Table 10 sensors-23-09085-t010:** Parametric estimation results of the metaheuristics: AOA, GA, BA, and MTDE.

-	${\hat{\Gamma }}_{\mathrm{AOA}}^{+}$	${\hat{\Gamma }}_{\mathrm{GA}}^{+}$	${\hat{\Gamma }}_{\mathrm{BA}}^{+}$	${\hat{\Gamma }}_{\mathrm{MTDE}}^{+}$
$X_{\dot{u}}$	−37.04	−14.30	−15.02	−49.82
$X_{u}$	−0.00	−0.00	−0.00	−0.00
$X_{v}$	−0.00	−0.00	−15.01	0.00
$X_{r}$	−0.01	0.00	1.24	−0.00
$X_{|u|u}$	−23.93	−16.98	−16.61	−13.92
$Y_{\dot{v}}$	−17.08	−16.55	−0.53	−64.14
$Y_{u}$	0.01	−0.01	0.00	−0.00
$Y_{v}$	−0.00	−0.00	−0.00	−0.00
$Y_{r}$	0.01	0.00	0.00	0.00
$Y_{|v|v}$	−134.13	−180.58	−266.76	−533.08
$N_{\dot{r}}$	−5.63	-6.36	−21.95	−8.96
$N_{u}$	−13.57	0.01	0.04	0.00
$N_{v}$	−0.00	−0.00	−34.58	0.00
$N_{r}$	−1.30	−9.89	−23.96	−0.01
N|r|r	−0.02	−1.46	−0.23	−1.77
${\bar{X}}_{\dot{v}}$	0.00	−0.00	−0.00	0.00
${\bar{X}}_{\dot{r}}$	−0.00	0.00	0.02	−0.00
${\bar{Y}}_{\dot{u}}$	0.00	−0.00	−0.00	0.00
${\bar{Y}}_{\dot{r}}$	−0.05	−0.01	−0.16	−21.30
${\bar{N}}_{\dot{u}}$	−0.00	−0.01	−0.00	0.14
${\bar{N}}_{\dot{v}}$	0.02	−0.01	−0.03	0.00
${\bar{X}}_{|u|v}$	−0.00	−0.00	−0.22	0.00
${\bar{X}}_{|u|r}$	−0.01	0.00	13.37	−0.00
${\bar{Y}}_{|v|u}$	0.01	−0.00	0.00	−0.00
${\bar{Y}}_{|v|r}$	0.00	0.00	0.00	0.00
${\bar{Y}}_{|r|u}$	−8.29	0.01	1.52	0.00
${\bar{Y}}_{|r|v}$	−0.01	0.00	0.00	0.02

**Table 11 sensors-23-09085-t011:** RMSE results for tracks 1, 2, and 3.

	C1	C2	C3	Total	Ranking
GWO	0.59	0.32	0.45	1.36	1
WOA	0.57	0.46	0.53	1.57	5
PSO	0.44	0.52	0.48	1.45	3
SSA	0.49	0.47	0.52	1.48	4
AOA	3.38	2.79	2.06	8.23	8
GA	0.75	0.44	0.59	1.77	6
BA	0.52	0.39	0.97	1.88	7
MTDE	0.49	0.48	0.47	1.44	2

**Table 12 sensors-23-09085-t012:** Analysis of Root-Mean Square Error (RMSE) results for each Degree of Freedom (DoF).

	*u* [m/s]	*v* [m/s]	*r* [rad/s]	Ranking
GWO	0.24	0.20	0.34	1 (Best)
WOA	0.26	0.29	0.35	5
PSO	0.23	0.26	0.34	3
SSA	0.36	0.21	0.28	4
AOA	1.29	0.35	2.21	8 (worst)
GA	0.28	0.31	0.43	6
BA	0.66	0.24	0.33	7
MTDE	0.26	0.27	0.30	2

**Table 13 sensors-23-09085-t013:** Qualitative analysis of algorithm performance—Validation tests (C1, C2, C3).

		RMSE of Each DoF			Total RMSE by Track				
	**SN (Signals Number Used in Model Desined)**	u **[m/s]**	v **[m/s]**	r **[rad/s]**	**T1**	**T2**	**T3**	**Total**	**Ranking**
GWO	5	0.24	0.20	0.34	0.59	0.32	0.45	1.36	1
WOA	3	0.26	0.29	0.35	0.57	0.46	0.53	1.57	5
PSO	5	0.23	0.26	0.34	0.44	0.52	0.48	1.45	3
SSA	4	0.36	0.21	0.28	0.49	0.47	0.52	1.48	4
AOA	1	1.29	0.35	2.21	3.38	2.79	2.06	8.23	8
GA	1	0.28	0.31	0.43	0.75	0.44	0.59	1.77	6
BA	4	0.66	0.24	0.33	0.52	0.39	0.97	1.88	7
MTDE	4	0.26	0.27	0.30	0.49	0.48	0.47	1.44	2

## Data Availability

Not applicable.

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
