# Peer review of "Performance Comparison of Meta-Heuristics Applied to Optimal Signal Design for Parameter Identification"

_sensors, 2023, doi:10.3390/s23229085_

Round 1
Reviewer 1 Report
Dear Authors,
Unfortunately, the novelty of the paper is unclear to me. The concept of rSOESGOPE (Section 3) is already presented in [22] (Neto et al.). It is based on PSO; in this paper you compare the use of established algorithms PSO, GWO, AOA, SSA, etc to create an "optimal" signal for parameter estimation, as described in Section 4. The comparison, when applied to the model of a and to the actual vessel, is presented in Section 5. This application was, once again, presented before [22], admittedly only for PSO. So, one could argue that the goal of the paper is to determine which algorithm is the best one. This is not really or, at least, not well discussed in the conclusions; some comments are made in Section 5.4. I also doubt that the results are transferable. In summary, unfortunately, I do not see much merit in the work and, for this reason, cannot recommend it for publication.
Sincerely
Further comments: Figures 5 to 12 should be made into one. Why are Figures 13 to 16 at the very end of the paper after the list with references and what are they good for?
While understable most of the time, the text is riddled with grammatical errors and sentences are badly structured. Moreover, the formatting is very often off and sloppy: E.g., lines go over into the margins and referencing styles are mixed; they are given either as numerals or last names of authors. Repeatedly, I find statements whose meanings elude me, such as: "It is a fact that, in principle, unplanned signals cannot be guaranteed." Finally, I also believe that information is missing in the affiliation part.
Author Response
Dear reviewer, thank you for your sincere words. The article proposes to investigate the application of new algorithms in the rSOESGOPE concept, which had not yet been carried out by (Neto et al., 2021). This is the main contribution of the article, which is not found in the literature either. Based on your remarks, we’ve improved the summary of contributions to make the intent of the article really clear.

Reviewer 2 Report
1. Summary: The study was conducted to evaluate the effectiveness of various meta-heuristics in Optimal Signal Design (OSD) for estimating parameters in nonlinear systems. The rSOESGOPE concept was utilized, and a catamaran-type Autonomous Surface Vessel (ASV) served as the practical case for the study. Nonetheless, the study affirmed the effectiveness of the rSOESGOPE concept across several algorithms, facilitating the creation of robust control strategies and fault-tolerant systems for ASVs operating in challenging environments.
2. Suggestions for Improvement and Future Research:
(i) A more in-depth analysis of the AOA method is recommended to comprehend its limitations better and enhance its performance.
(ii) A comprehensive assessment of the algorithms' performance under varying environmental conditions and real-world scenarios would offer a more holistic understanding of their applicability and limitations.
(iii) Creating a hybrid meta-heuristic model that combines the strengths of the top-performing algorithms could lead to a more robust and efficient solution.
Author Response
Dear reviewer, we really appreciate your time reviewing our paper. We hope we reached your expectations.

Round 2
Reviewer 1 Report
Accept after minor revision
Accept after minor revision
Author Response
In the foremost place, we want to extend our heartfelt gratitude for the generous allocation of your time and your unwavering dedication to enhancing our work. Your consistent efforts have been instrumental in not only improving the quality of our projects but also strengthening our team's collaborative spirit. Your contributions are truly invaluable, and we deeply appreciate your commitment.
We would like to emphasize that we engaged the services of a highly regarded English professional to meticulously review the entirety of the article's content. Thanks to their expert revision, we are confident in presenting it to you once more with substantial enhancements in terms of English grammar. The knowledge and expertise brought by this specialist have played an instrumental role in elevating the quality of our work, ensuring it meets the highest linguistic standards.
The changes made to the text are minor and do not alter the technical context of the work. Therefore, we believe it is unnecessary to highlight each intervention in red. The reviewed version is here attached.
